# ExDBN: Exact learning of Dynamic Bayesian Networks

## Abstract

Causal learning from data has received a lot of attention in recent years. One way of capturing causal relationships is by utilizing Bayesian networks. There, one recovers a weighted directed acyclic graph in which random variables are represented by vertices, and the weights associated with each edge represent the strengths of the causal relationships between them.

This concept is extended to capture dynamic effects by introducing a dependency on past data, which may be captured by the structural equation model. This formalism is utilized in the present contribution to propose a score-based learning algorithm. A mixed-integer quadratic program is formulated and an algorithmic solution proposed, in which the pre-generation of exponentially many acyclicity constraints is avoided by utilizing the so-called branch-and-cut ("lazy constraint") method.

Comparing the novel approach to the state-of-the-art, we show that the proposed approach turns out to produce more accurate results when applied to small and medium-sized synthetic instances containing up to 25 time series. Lastly, two interesting applications in bioscience and finance, to which the method is directly applied, further stress the importance of developing highly accurate, globally convergent solvers that can handle instances of modest size.

## 1 Introduction

The problem of causal learning using graphical structures has received considerable attention from a wide range of communities in recent years. This attention comes from the wide range of applications including, but not limited to, medicine (Rajapakse & Zhou, 2007), machine learning (Koller & Friedman, 2009), econometrics (Luetkepohl, 2005; Demiralp & Hoover, 2003; Malinsky & Spirtes, 2018) and others (Guo et al., 2020; Assaad et al., 2022).

One key reason for this is that in many applications data is abundant, but modeling using first principles may be difficult due to the complexity of the problem at hand (Guo et al., 2020). Some of this complexity may arise due to an abundance of non-linear effects, only a partial ability to observe the system, or unexpected stochastic effects influencing the system. For a detailed discussion on these, please refer to Friedman et al. (2013); Kungurtsev et al. (2024).

Other issues that are inherent to graphical structure learning from time series data are related to the sampling timescales and scaling to large instances, these have been addressed in Abavisani et al. (2023); Ouyang et al. (2024), respectively. A key benefit of learning via graphical structures is the full explainability of the output; the network may be either used to compute outputs for different situations or the learned graph structure may be inspected and dependencies of particular interest analyzed.

In this contribution, we revisit the score-based learning of dynamic Bayesian networks utilizing a directed acyclic graph (DAG) structure augmented by additional time dependencies from data (Murphy, 2002; Dean & Kanazawa, 1989; Assaad et al., 2022). This approach to learning causality has been successfully applied to a variety of problems, many of which are related to applications in medicine (Zandonà et al., 2019; van Gerven et al., 2008; Michoel & Zhang, 2023; Zhong et al., 2023). In addition to medical applications, the dynamic Bayesian network approach representations are widely used in econometrics (Hoover & Demiralp, 2003b) and financial risk modeling (Ballester

et al., 2023). This broad scope of applications has spawned a large number of excellent solvers that, under different assumptions, can discover the underlying causal structure of a system. The use of various assumptions is key to ensure the tractability of a solver, since the the number of constraints that is needed to impose to acyclicity of the representing graph is super-exponential in the number of random variables.

One of the possible assumptions is to separate observational and interventional data (Gao et al., 2022), which reduces the number of dependencies that need to be found. Another is the assumption of continuous underlying dynamics represented by stochastic differential equations Bellot et al. (2021). One can also assume a priori knowledge about time-lagged data and incorporate this knowledge into the solver Sun et al. (2021). One can also deal with the general problem and propose local methods (Pamfil et al., 2020; Gao et al., 2022), which can scale further at the cost of some loss of accuracy. Note that many of the previous works also combine several of these approaches to arrive at solvers that are tractable and applicable to a wide range of applications. However, it should be noted that many methods may not identify DAG representations of casual dependencies under certain conditions Kaiser & Sipos (2022); Reisach et al. (2021). One of the possible causes is that many of them only converge to a local stationary point for the optimization problem.

We utilize mixed-integer programming to learn dynamic Bayesian networks. All of the previous works mentioned above focus mostly on solving the curse of dimensionality and scaling with adequate precision. On the other hand, we focus on leveraging fundamental principles that apply to quadratic mixed-integer programs to find global solutions to the score-based DAG learning problem, which results in a high-quality reconstruction of the DAG. Furthermore, we tackle the curse of dimensionality by avoiding the pre-generation of the acyclic constraints. It is shown that given sufficient data, only a small amount of these constraints are actually needed to ensure the acyclicity of the resulting graph, which leads to the runtime generation of these constraints granting a large speedup over the version of the algorithm that uses all of the constraints for the entire duration of the computation. The formulation and its implementation are easily reproducible, making it accessible to a wide range of potential practitioners.

## 2 PROBLEM FORMULATION

Before formulating the problem of score-based Bayesian network learning as a mixed-integer program, let us describe the state space using a structural vector autoregressive model (Hoover & Demiralp, 2003a; Kilian, 2011). Let $d, T \in \mathbb{N}$ and assume that $X_{i,t}$ is a set of stochastic processes, where $i \in \{1, 2, \ldots, d\}$ and $t \in \{1, 2, \ldots, T\}$. Let the underlying DAG to be learned be characterized by the set of vertices and edges organized in a pair $(V, E)$, where the vertices are indexed by the set of integers $\{1, 2, \ldots, d\}$ and $E \subset V \times V$. Denote the auto-regressive order by $p \in \mathbb{N}$ and let

$$W \in \mathbb{R}^{d,d}, \quad A_i \in \mathbb{R}^{d,d}, \quad i \in \{1, 2, \ldots, p\}, \tag{1}$$

be the weighed adjacency matrix of $(V, E)$ and $A_i$ be the matrices encoding the time regressive dependencies. The intra-slice interactions defined at the present time are expressed by the weight matrix $W$ and the inter-slice interactions are expressed by $A_i$. For simplicity, the matrices $A_i$ are assumed to be constant. Let $X_t \in \mathbb{R}^{n,d}$ be the data matrix at time $t$, then the linear auto-regressive model of order $p$ reads

$$X_t = X_t W + X_{t-1} A_1 + X_{t-2} A_2 + \ldots + X_{t-p} A_p + Z, \tag{2}$$

where $Z \in \mathbb{R}^{n,d}$ is the error vector, which is not assumed to be Gaussian. Note that non-linear autoregressive models can also be formulated in an analogous way. The problem may be written in a simplified manner as

$$X_t = X_t W + Y_t A + Z, \tag{3}$$

where

$$A = A_1 | A_2 \ldots | A_p, \quad Y_t = X_{t-1} | X_{t-2} \ldots | X_{t-p}. \tag{4}$$

To maximize the fit of the data over the model, a score function, which reads may be formulated

$$J(W, A) = \|X - XW - YA\|_F^2 + \lambda \|W\| + \eta \|A\|, \tag{5}$$

where $\|\cdot\|$ denotes an arbitrary matrix norm and $\lambda, \eta > 0$ are sufficiently small regularization coefficients. The problem of interest then reads

$$
\min_{W,A} J\left(W, A\right),
$$
$$
G\left(W\right) \in \Gamma_{DAG}, \tag{6}
$$

where $A$ need not be constrained, since cycles are excluded by construction; $\|\cdot\|$ denotes an arbitrary norm, which is usually chosen to be the L1-norm and $\|\cdot\|_{\mathbb{F}}$ denotes the Frobenius norm.

**Remark 1** *The identifiability of $W$ and $A$ using 6 has been studied for Gaussian and non-Gaussian noise. Regardless of noise, the identifiability of $A$ is a consequence of the basic theory of autoregressive models (Kilian, 2011). The identifiability of $W$ is a bit more involved and must be separated into the Gaussian and non-Gaussian case. However, in either case, identifiability is possible under mild conditions (Hyvärinen et al., 2010; Peters & Bühlmann, 2012).*

## 3 BRIEF INTRODUCTION TO MIXED INTEGER QUADRATIC PROGRAMMING

To better frame the content of Section 4, we provide a short introduction to mixed-integer quadratic programming. An optimization problem, is called a mixed-integer quadratically constrained quadratic program (MIQCQP) if it is of the form

$$
\min_{x \in \mathbb{R}^n} \quad x^T Q x + q^T x, \tag{7}
$$
$$
\text{s.t.} \ x^T Q_i x + q_i^T x \le a_i, \tag{8}
$$
$$
Ax \le b, \tag{9}
$$
$$
x \in F \tag{10}
$$

where $Q, Q_i \in \mathbb{R}^{n,n}$, $q, q_i \in \mathbb{R}^n$, $A \in \mathbb{R}^{m,n}$, $a \in \mathbb{R}^k$, $b \in \mathbb{R}^m$, $F$ is a product of the form

$$
F = \underbrace{\mathbb{R} \times \ldots \times \mathbb{R}}_{n-r \text{ times}} \times \underbrace{\mathbb{N} \times \ldots \times \mathbb{N}}_{r \text{ times}} \tag{11}
$$

and $m, n, k, r \in \mathbb{N}$. Equation equation 7 is often called the cost or loss function, equation 8 represents the quadratic constraints, equation 9 are the linear constraints, and $F$ is the set that enforces the integrality constraints for the $r$ components of the decision variable $x$.

Mixed-integer quadratic programs have been shown to be in NP Del Pia et al. (2014), which often leads to an exhaustive demand for computational resources. The algorithms used to solve MIQP are typically branch-and-bound or cutting plane Dakin (1965); Bonami et al. (2009); Westerlund & Pettersson (1995); Kronqvist et al. (2015). Both of these algorithmic treatments are often employed together, often with the addition of a presolving step, the use of heuristics and parallelism. The aforementioned allows many modern solvers to solve even large problems despite the NP hardness. Some of these solvers are open source (like SCIP and GLPK) and others are commercial (GUROBI and CPLEX). The powerful infrastructure present in these solvers can be made use of together with additional problem-specific modifications to deliver high-quality solutions.

Due to the exhaustive nature of the algorithms mentioned in the previous paragraph, global convergence is guaranteed Belotti et al. (2013). Furthermore, convergence to the global solution may be tracked and the error estimated by computing the dual problem of (7–10). The dual of the problem is then used to computed the so called MIP GAP as follows

$$
\text{MIP GAP} = \frac{|J\left(x^*\right) - J_{\text{dual}}\left(y^*\right)|}{|J\left(x^*\right)|}, \tag{12}
$$

where $x^*$ and $y^*$ are the current best solutions of the primal and dual problems respectively, and $J$ and $J^*$ are the cost functions of the primal and dual problems, respectively. The MIP GAP ensures that we can assess the quality of the minimization during solution time and terminate the computation when the result is good enough (small enough MIP GAP). Furthermore, if the gap reaches 0 at any point, we are sure that the current solution is a global optimum.

## 4 MIXED INTEGER QUADRATIC PROGRAMMING FORMULATION

Formulating the learning problem as a mixed-integer quadratic problem sets things up so that a globally convergent algorithm may be used. This is fundamental for high-precision learning to be possible.

Let $e_{i,j} \in \{0,1\}$ and $e_{i,j}^s \in \{0,1\}$ be decision variables that govern the placement of edges between random variables at time level $t$ and between time levels $t$ and $t-s$, respectively, and let $w_{i,j} \in \mathbb{R}$ and $a_{i,j}^t \in \mathbb{R}$ be the associated edge weights. Using these variables, the scoring function of problem equation 6 becomes

$$J_p = \sum_{i=1}^n \sum_{j=1}^d \left| X_{i,j} - \sum_{k=1}^d X_{i,k} w_{k,j} - \sum_{s=1}^p \sum_{k=1}^d X_{i,k}^s a_{k,j}^s \right|^2 + \text{REG}, \tag{13}$$

which avoids the use of a bi-linear term if the additional constraints

$$w_{k,j} \leq c e_{k,j}, \quad w_{k,j} \geq -c e_{k,j} \text{ for all } k, j \in \{1, 2, \ldots, d\}. \tag{14}$$

and

$$a_{k,j}^s \leq c e_{k,j}^s, \quad a_{k,j}^s \geq -c e_{k,j}^s \text{ for all } k, j \in \{1, 2, \ldots, d\}, s \in \{1, 2, \ldots, p\} \tag{15}$$

are imposed, where $c > 0$ is the maximal admissible magnitude of any weight and $\lambda > 0$ is a regularization constant. Note that the maximal admissible regularization is chosen so as not to affect the result of the identification, i.e. $c = 100$, but the true edge weights are two orders of magnitude smaller.

Where REG is a regularization expression equals either: (L1)

$$\text{REG} = \lambda \sum_{i=1}^n \sum_{j=1}^n e_{i,j} + \eta \sum_{s=1}^p \sum_{i=1}^n \sum_{j=1}^n e_{i,j}^s. \tag{16}$$

or (L2)

$$\text{REG} = \lambda \sum_{i=1}^n \sum_{j=1}^n e_{i,j} + \eta \sum_{s=1}^p \sum_{i=1}^n \sum_{j=1}^n a_{i,j}^s. \tag{17}$$

Lastly, the acyclicity constraints are described. Let $C$ denote the set of all cycles in a graph with $d$ vertices, where each cycle $c \in C$ of length $k$ is represented as a set of edges: $c = \{(i_1, i_2), (i_2, i_3), \ldots, (i_{k-1}, i_1)\}$. The constraint excluding a cycle $c \in C$ from a solution then reads

$$\sum_{(i,j) \in c} e_{i,j} \leq k - 1. \tag{18}$$

The algorithmic treatment of constraint equation 18 is key in the following section, in which the algorithmic treatment is discussed as implementing the branch-and-bound-and-cut algorithm without a reduction mechanism for this constraint is doomed to fail due to the super-exponential number of such constraints.

## 5 ALGORITHMIC IMPLEMENTATION USING BRANCH-AND-BOUND-AND-CUT

One of our main contributions is the development of a branch-and-bound-and-cut algorithm to solve the formulation mentioned above. Since the acyclic constraints 18 need to be imposed only for the edges of the graph representing the intra-slice level, all of what follows is only applied to the intra-slice graph. While we leverage the traditional branch-and-bound approach as described in (Achterberg, 2007, e.g.), we incorporate cycle exclusion constraints equation 18 using "lazy" constraints. These are only enforced once an integer-feasible solution candidate is found. If a violation of a lazy constraint occurs, the constraint is added across all nodes in the branch-and-bound tree. At the root node, only $O(|E|)$ constraints 14 and 15 are initially used. Cycle-exclusion constraints equation 18 are added later. Note that this method is not a heuristic and does not lead to a possibly harmful reduction (or extension) of the solution space leading to omitting possible solutions or

returning solutions which are not DAGs. Furthermore, it is shown that the number of constraints that are actually needed in a computation is many orders of magnitude less than the number of all possible constraints.

Once a new mixed-integer feasible solution candidate is identified, detecting cycles becomes straightforward using a depth-first search (DFS). If a cycle is detected, the corresponding lazy constraint equation 18 is added to the problem. The DFS algorithm solves the problem of cycle detection in a worst-case quadratic runtime relative to the number of vertices in the graph, which contrasts with algorithms that separate related inequalities from a continuous relaxation (Borndörfer et al., 2020; Cook et al., 2011), such as the quadratic program in our case. Three variants of adding lazy constraints for the problem were tested.

- Adding a lazy constraint only for the first cycle found.
- Adding a lazy constraint only for the shortest cycle found.
- Adding multiple lazy constraints for all cycles found in the current iteration in which an integer-feasible solution candidate is available.

The third mentioned variant was found to consistently deliver the best results, despite (Achterberg, 2007, Chapter 8.9). Therefore, it is applied in all the numerical tests that follow.

## 6 DATA GENERATION

We generate data in a manner similar to that described in Zheng et al. (2018) and Pamfil et al. (2020). The evaluation of ExDBN was performed on the synthetic data generated as follows. First, a random intra-slice DAG was created using either the Erdős-Rény (ER) model or the scale-free Barabási–Albert (SF) model. The DAG weights were sampled uniformly from the intervals $[-2.0, -0.5] \cup [0.5, 2.0]$.

Next, inter-slice graphs were generated using the ER model. For each inter-slice graph, weights were sampled from the interval $[-0.5\alpha, -0.2\alpha] \cup [0.2\alpha, 0.5\alpha]$, where $\alpha = 1/\eta^{t-1}$, $\eta \geq 1$ is the decay parameter, and $t$ is the time of the slice. $t = 0$ corresponds to the intra-slice, while $t \in \{1, \ldots, p\}$ represents the inter-slices.

The data samples are then generated using the structural equation model equation and adding Gaussian noise with either variance 1 or different variance for each variable sampled uniformly from a given interval.

Specifically, we have adapted the ER and SF generators from Zheng et al. (2018) for dynamic networks. Notice that this results in a slightly different generator than in Pamfil et al. (2020), which may explain some of the differences in the performance of DYNOTEARS, compared to the original article.

## 7 NUMERICAL EXPERIMENTS

In recent years, many solvers have been developed to facilitate the graphical learning of Bayesian networks that represent causality (Pamfil et al., 2020; Hyvärinen et al., 2010; Malinsky & Spirtes, 2018; Gao et al., 2022; Dallakyan, 2023; Lorch et al., 2021). Each of these solvers (including the one presented) faces the curse of dimensionality, which somewhat restricts the applicability of each solver, and thus through testing needs to be provided. It is impossible to test the proposed solution w.r.t. every solver developed. There is, however, a significant branch of development that allows for direct comparison, and by transitivity of results, the comparison with many previous solvers follows.

In 2020, Pamfil et al. (2020) have developed a locally convergent method, called DYNOTEARS, that learns causality as a Bayesian network that supersedes the solution methods previously developed (Hyvärinen et al., 2010; Malinsky & Spirtes, 2018; Zheng et al., 2018). Further developments based on previous publications include formulating the problem in the frequency domain or defining differentiable Bayesian structures (Dallakyan, 2023; Lorch et al., 2021). In the following, we provide a head-to-head comparison with DYNOTEARS and thus by transitivity with the methods documented by Hyvärinen et al. (2010); Malinsky & Spirtes (2018).

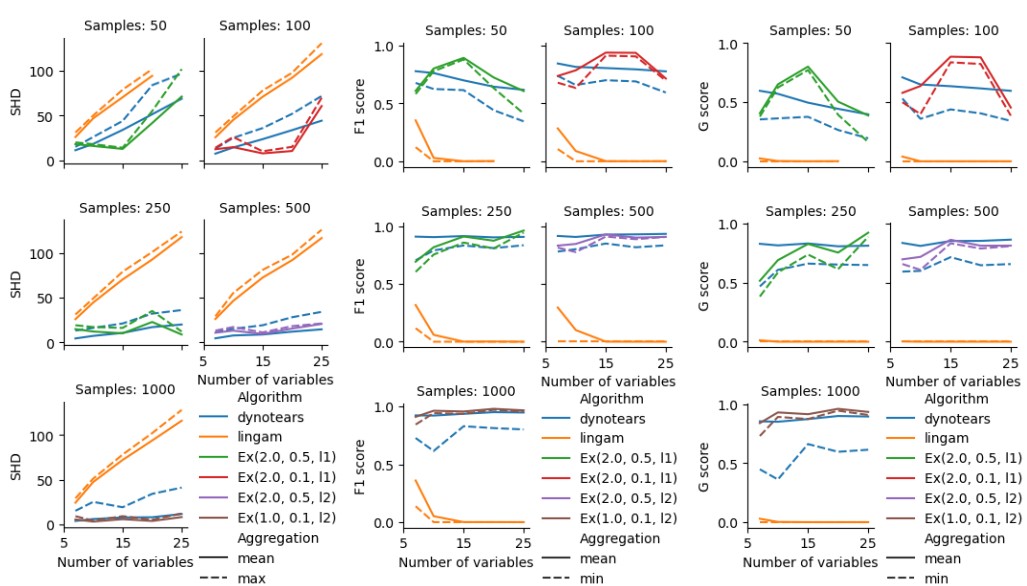

Figure 1: SHD, F1 score, and G score for a test case using ER3-1 random ensemble, i.e., edge-vertex ratio 3 on intra graph, edge-vertex ratio 1 on inter graph, recursion depth 1. Variance of noise is equal to 1 for all variables. Ex($\lambda$, $\eta$, l1) means ExDBN algorithm with L1 regularization and coefficients $\lambda$, $\eta$.

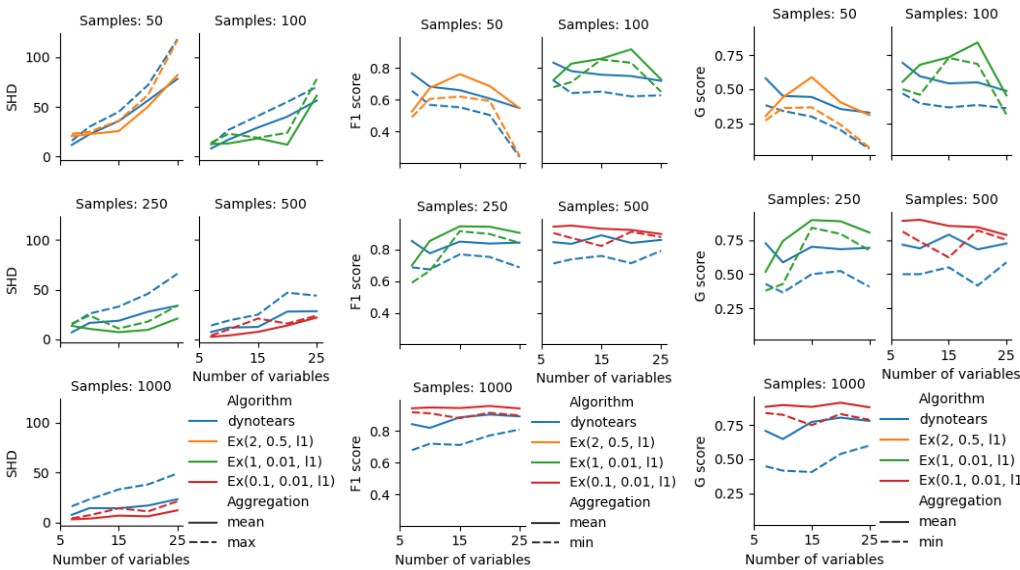

Figure 2: SHD, F1 score, and G score for a test case using ER3-1 random ensemble, i.e., edge-vertex ratio 3 on intra graph, edge-vertex ratio 1 on inter graph, recursion depth 1. Variance of noise is randomly sampled from uniform distribution on interval $(0.6, 1.2)$ for each variable.

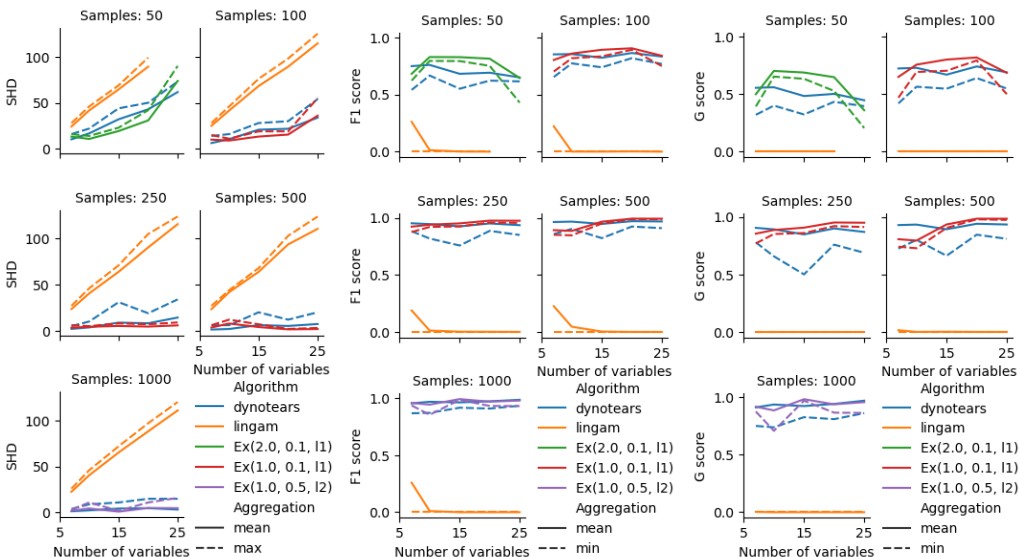

Figure 3: SHD, F1 score, and G score for a test case using SF3-1 random ensemble, i.e., edge-vertex ratio 3 on intra graph, edge-vertex ratio 1 on inter graph, recursion depth 1. Variance of noise is equal to 1 for all variables.

## 7.1 BENCHMARK SETUP AND QUANTITIES OF INTEREST

In Section 7.2, $W_{\text{true}}$ denotes the adjacency matrix representing the intra-slice dependencies and $A_{\text{true}}$ denotes the inter-slice dependencies of the ground truth, where $A_{\text{true}}$ is used to denote a $p$-tuple as in equation 4. $W_{\text{true}}$ and $A_{\text{true}}$ are used to generate data while applying Gaussian distribution noise. Following the data generation process, the matrices $W$ and $A$ are identified and compared with $W_{\text{true}}$ and $A_{\text{true}}$. Because noisy data inevitably leads to some falsely identified edges, typically with negligible weights, edges with a weight less than $\delta > 0$ can be removed from $W$ and $A$, resulting in a graph $W^{\delta}$ and $A^{\delta}$, respectively. To compare methods for known ground truth $W_{\text{true}}$ and $A_{\text{true}}$, we choose the best possible $\delta > 0$ for each method. This $\delta > 0$ may then be used as a reference for learning from data, where a ground truth is not known. Next, we introduce the relevant metrics used to evaluate the quality of the reconstruction, when $W_{\text{true}}$ is available.

In the following, we suppose that a DBN represented by an inter-slice matrix $V$ and an inter-slice matrix $A$ is denoted by an ordered pair $(V, A)$. Let $(V, A)$ and $(V, A)$ be two such pairs, then one defines the structural Hamming distance (SHD) as

$$\rho(V, A; W, B) = \sum_{i,j=1}^{d} r_{ij}(V, W) + \sum_{k=1}^{p} \sum_{i,j=1}^{d} r_{ij}(A_k, B_k), \tag{19}$$

where

$$r_{ij}(C, D) = \begin{cases} 0 & \text{if } C_{ij} \neq 0 \text{ and } D_{ij} \neq 0 \text{ or } C_{ij} = 0 \text{ and } D_{ij} = 0 \\ \frac{1}{2} & \text{if } C_{ij} \neq 0 \text{ and } D_{ji} \neq 0 \\ 1 & \text{otherwise.} \end{cases} \tag{20}$$

SHD is used as a score that describes the structural similarity of two DAGs in terms of edge placement and is commonly used to assess the quality of solutions (Zheng et al., 2018; Pamfil et al., 2020). Besides SHD,

$$\text{precision} = \frac{\text{true positive}}{\text{true positive} + \text{false positive}}, \quad \text{and} \quad \text{recall} = \frac{\text{true positive}}{\text{true positive} + \text{false negative}}, \tag{21}$$

are used Andrews et al. (2024) to evaluate the quality of structural recovery. It is important to note that precision and recall isolate false positives and negatives, respectively, in contrast to SHD, where these quantities are both accounted for simultaneously. The last metric that can be used to evaluate

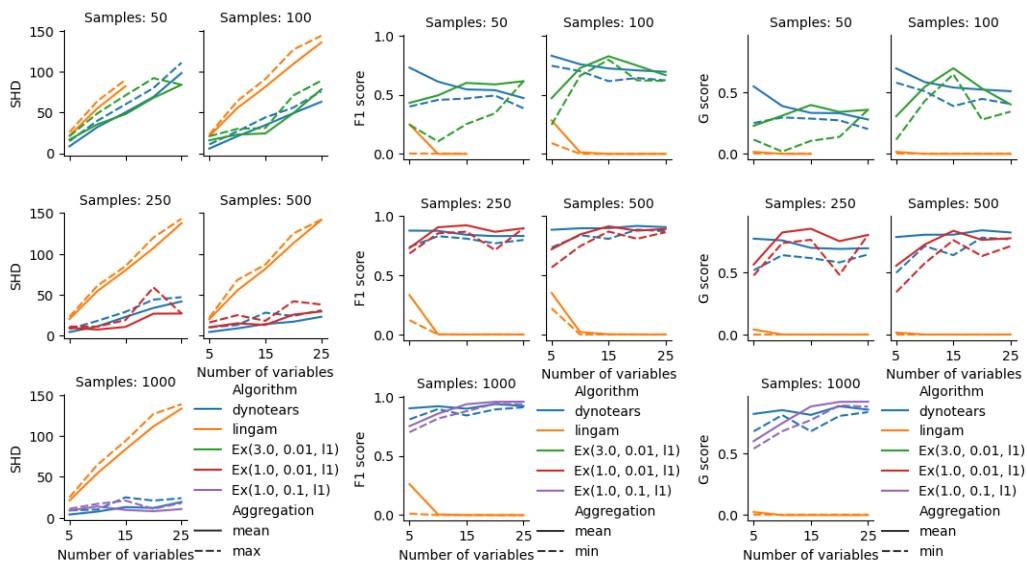

Figure 4: SHD, F1 score, and G score for a test case using ER2-1-1 random ensemble, i.e., edge-vertex ratio 2 on intra graph, edge-vertex ratio 1 on inter graphs, recursion depth 2. Variance of noise is equal to 1 for all variables.

structural similarity is the F1 score and reads

$$F_1 = \frac{2}{\text{precision}^{-1} + \text{recall}^{-1}}. \tag{22}$$

Note that all of the quantities evaluated in equation 21 and equation 22 are a result of summing up all of the differences over both inter- and intra-slice dependencies between a given pair $(V, W)$ and a ground truth.

Although structural similarity is a key concern, merely comparing structural properties does not tell the full story, as the weights play a crucial role in the resulting statistical behavior of the found DAG. This motivates us to additionally utilize a cost function based metric, which reads

$$\sigma_p(V, W) = |J_p(V) - J_p(W)|, \tag{23}$$

where $\lambda = 0$ and typically $p = 2$. We may also evaluate the differences in adjacency matrices by considering

$$\|V - W\|_{\mathbb{F}}, \tag{24}$$

where $\|\cdot\|_{\mathbb{F}}$ denotes the Frobenius norm.

### 7.2 SYNTHETIC BENCHMARK RESULTS

In the following benchmark, the generation methods described in Section 6 are used to compare ExDBN with DYNOTEARS (Pamfil et al., 2020) under the assumption of Gaussian noise. Even though the cost function is a maximum likelihood estimator (see Section 1) for non-Gaussian noise, we leave this evaluation for future publication. The scaling is studied for different numbers of variables, samples, and graph generation methods with the relevant metrics; SHD, F1 score, and G score recorded in Figures 1, 2, 3 and 4.

A statistical ensemble with 10 different seeds was used for each of the experiments, and the mean and worst possible case values are used in the plots. It should be noted that, naturally, the worst possible value and the mean can be used together to bound the variance. The solution time is capped for ExDBN at 7200 seconds, and the regularization applied in ExDBN needs to be scaled appropriately with the number of samples, as it is assumed that the optimal choice of regularization constant is a decreasing function of the number of samples. We use the aforementioned as a guide

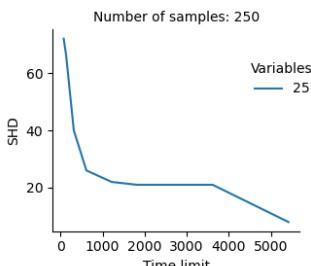

Figure 5: Comparison of ExDBN solution quality (SHD) versus running time on SF3-1 problem.

(in a nonstrict way) to find the right regularization for a given sample size. This follows from the fact that the regularization is to be kept proportionally small to the main objective expressed by equation 13. Furthermore, it was found that changing the regularization from L1 to L2 is beneficial for identification when the number of samples is large. Furthermore, if we do not know the ground-truth graph. We can try to run the algorithm for multiple values of $\lambda$ and $\eta$ and then use the one that produces a better MIP GAP. For a smaller number of samples, L1 regularization works better. For a larger number of samples, L2 yields good results and is usually faster.

As noted in (Reisach et al., 2021), the noise variances and data scale may be important for some algorithms to perform well. We tested ExDBN on normalized data and noticed a significant performance drop. Therefore, ExDBN is suitable for problems in which the data of the samples have a true scale.

The results of the tests can be divided into two categories by the average number of edges. Figures 1, 2 and 3 show higher-degree graphs (average degree 3) and Figure 4 shows the reconstruction of a lower-degree graph. In the case of the lower degree graph, it is clear that both DYNOTEARS and ExDBN perform similarly, with ExDBN performing better than DYNOTEARS some of the time, with the converse being true equally often.

In the case of the identification of higher degree graphs, however, one can notice that the worst possible performance and the mean performance are much closer in the case of ExDBN, where we can point out for instance the G score in the case with 1000 samples. In these instances, the difference between the worst possible G score difference is between 0.3 and 0.5 in the case of DYNOTEARS but stays well below 0.1 in the case of ExDBN. The aforementioned can be interpreted as superior reliability of the solution as the worst possible reconstruction is consistently better.

Focusing on the 1000-sample case, while somewhat taking into account the previous ones, too, we see that the performance gap between the solvers increases in favor of ExDBN as we increase the number of samples. In the lower sample cases, one may also observe that ExDBN outperforms DYNOTEARS for many graph sizes in the mean and consistently outperforms DYNOTEARS in the worst possible case (min/max depending on the metric).

Note that the global convergence of the method, which is rooted in the fundamentals of mixed-integer quadratic programming, allows us to increase the computation time, which leads to improving the metrics reported further. While some time-sensitive applications like short-term stock evaluation might not be able to benefit from this, others like biomedical applications might benefit as a computation lasting several days, in which the accuracy in measurably improved (by monitoring the duality gap) is desirable. See Figure 5 for the comparison of running time versus solution quality.

We also made a comparison with VarLiNGAM (Hyvärinen et al., 2010). We used the default settings of the algorithm. ExDBN performed better in all scenarios tested.

### 7.3 APPLICATION IN BIOMEDICAL SCIENCES

In biomedical sciences, there is a keen interest in learning dynamic Bayesian networks to estimate causal effects (Tennant et al., 2020) and identify confounding variables that require conditioning. A recent meta-analysis (Tennant et al., 2020) of 234 articles on learning DAGs in biomedical sciences

found that the averaged DAG had 12 nodes (range: 3–28) and 29 arcs (range: 3–99). Interestingly, none of the DAGs were as sparse as the commonly considered random ensembles; median saturation was 46%, that is, each of all possible arcs appeared with probability 46% and does not converge to a global minimum of the problem.

As an example, we consider a recently proposed benchmark of Ryšavý et al. (2024), where the Krebs cycle is to be reconstructed from time series of reactant concentrations of varying lengths. There, DYNOTEARS cannot reach the (Ryšavý et al., 2024) F1 score of 0.5 even with a very long time series. In contrast, our method can solve instances equation 13 to global optimality. Using ExDBN, however, the global minimization is ensured given sufficient time and thus the maximum likelihood estimator is found. However, it should be noted that depending on the number of samples and noise, it may be that even the maximum-likelihood estimator is not sufficiently accurate. However, this does not reflect poorly on the method itself, but is rather a matter of the modification of data collection methods associated with the experiment. In a one-hour time limit, ExDBN can find a solution with the 38% duality gap.

### 7.4 Application in Finance

In financial services, there are also several important applications. The original DYNOTEARS paper considered a model of diversification of investments in stocks based on dynamic Bayesian networks. Independently, Ballester et al. (2023) consider systemic credit risk, which is one of the most important concerns within the financial system, using dynamic Bayesian networks. They found that the transport and manufacturing sectors transmit risk to many other sectors, while the energy sector and banking receive risk from most other sectors. To a lesser extent, there is a risk transmission present between approximately 25% of the sectors pairs, and these network relationships explain between 5 % and 40 % single systemic risks. Notice that these instances are much denser than the commonly used random ensembles.

We elaborate on the example of Ballester et al. (2023), where 10 time series capture the spreads of 10 European credit default swaps (CDS). Considering the strict licensing terms of Refinitiv, the data from Ballester et al. (2023) are not available from the authors, but we have downloaded 16 time-series capturing the spreads of 16 European CDS with RED6 codes 05ABBF, 06DABK, 0H99B7, 2H6677, 2H66B7, 48DGFE, 6A516F, 8A87AG, 8B69AP, 8D8575, DD359M, EFAGG9, FF667M, FH49GG, GG6EBT, NN2A8G, from January 1st, 2007, to September 25th, 2024. This amounts to more than 11 MB of time series data when stored as comma-delimited values in plain text. Although the procedure for learning the dynamic Bayesian network in Ballester et al. (2023) is rather heuristic, we can solve the mixed integer programming (MIP) instance for the 16 European CDS in 2 minutes. In the heuristic of Ballester et al. (2023), they first perform unconditional independence tests on each set of two time series containing an original series and a lagged time series, to reduce the subsequent number of unconditional independence tests performed. There are 45 unconditional and conditional independence tests performed first, to suggest another 200 conditional independence tests. We stress that the procedure of Ballester et al. (2023) does not come with any guarantees, while our instance equation 13 is solved to global optimality. The run-time to global optimum of 2 minutes (using L2 regularization) validates the scalability of mixed-integer programming solvers.

## 8 Conclusion

Dynamic Bayesian networks have wide-ranging applications, including those in biomedical sciences and computational finance, as illustrated above. Unfortunately, their use has been somewhat limited by the lack of well-performing methods to learn them. Our method, ExDBN, provides the best possible estimate of the DBN, in the sense of minimizing empirical risk equation 13. Significantly, our method does not suffer much from the curse of dimensionality, even for real-world dense instances, which are typically challenging for other solvers. This is demonstrated most clearly in the case of systemic risk transmission detailed in Section 7.4, in which the global minimizer is found in 2 minutes. Additionally, the use of the guarantees on the distance to the global minimizer (so-called MIP gap, available ahead of the convergence to the global minimizer) provides a significant tool for fine-tuning the parameters of the solver in the case of real-world application, where the ground truth is not available. Combined with global convergence guarantees of the maximum likelihood estimator, this provides a robust method with state-of-the-art statistical performance.

DISCLAIMER

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
