# OpenReview forum: "ExDBN: Exact learning of Dynamic Bayesian Networks"
_ICLR.cc/2025/Conference — Submitted to ICLR 2025_

### Official Review · Reviewer_aH4X · 2024-10-27

**Soundness:** 3
**Presentation:** 3
**Contribution:** 2
**Rating:** 5
**Confidence:** 3

**Summary:**

A mixed-integer quadratic programming model is formulated, and an algorithmic solution is proposed that circumvents the need for pre-generating an exponential number of acyclicity constraints. This is achieved through the branch-and-cut (or 'lazy constraint') method, which also addresses the curse of dimensionality by dynamically handling the acyclic constraints during the solution process.

**Strengths:**

1- Structure learning, particularly for dynamic Bayesian networks and time series, is an essential first step in applying Bayesian inference. Therefore, the problem addressed in this paper holds significant practical value.

2- The numerical results for comparison accuracy are sufficient.

**Weaknesses:**

1- In general, convergence in mixed-integer programming cannot be guaranteed.  For the proposed algorithm which is based on mixed-integer programming, are there any theoretical guarantees or empirical evidence of convergence for their specific formulation? The soundness and correctness of the proposed algorithm need to be formally proven.

2- While the main idea is well presented, the authors have not clearly conveyed the details of the proposed method.

3- One of the key contributions of the proposed method appears to be reducing computational burden through the branch-and-cut (or 'lazy constraint') method. However, there is no comparison or analysis provided regarding this computational burden. As is well-known, the computational cost of mixed-integer programming is a significant challenge to its practical application. It would be better if authors compare the runtime of the proposed algorithm with existing methods or analyze the number of constraints generated during the solution process.

4- Several alternative methods for DBN structure learning were neither mentioned nor compared, such as constraint-based algorithms like PCMCI and PCMCI+, or noise-based approaches like TiMINo, VarLiNGAM, and oCSE. Additionally, methods like the Temporal Causal Discovery Framework (TCDF) leverage deep neural networks with attention mechanisms within dilated depthwise convolutional networks to learn complex nonlinear causal relationships between time series. Why do authors only choose the DYNOTEARS method for comparison?  Comparison with some other methods could indicate the improvement of the proposed method on the numerical results.

5- The authors could use statistical tests, such as the Wilcoxon signed-rank test, to compare numerical results.

6-There are some typographical errors. For instance, some variables, such as $\Gamma_{DAG}$ ​ in (6), are not defined. Additionally, the variable $c$ is used to represent two different quantities in (9) and (12).

**Questions:**

1- How can we select the parameters $\lambda$, $\eta$, and REG?

2- What is the termination condition for the proposed algorithm, and how can we appropriately determine it?

3- How does the proposed method differ from or improve upon existing techniques for learning dynamic Bayesian networks, and what specific challenges their method addresses that previous approaches could not? Please, clarify your contributions.

---

> ### Author Response · Authors · 2024-11-22
>
> We thank the reviewer for their insightful remarks and comments. We would, however, like to give the following responses:
> 1. “ In general, convergence in mixed-integer programming cannot be guaranteed…” - Using the branch and bound and cut method guarantees global convergence in general (basic theory, see here for instance: https://www.sciencedirect.com/topics/computer-science/branch-and-cut), typically at the cost of scalability. We deal with the scalability issue in the article (see Sections 2 through 4).
> 2. “While the main idea is well presented, the authors have not clearly conveyed the details of the proposed method.” - Would you please clarify which details would be of interest?
> 3. “One of the key contributions of the proposed method appears to...” - The potential number of constraints is  equal to the number of directed cycles in a complete graph on n vertices, which is $\sum_{i=1}^n \binom{n}{i}(i-1)!$. Due to this count being very large, the examples we present, would not fit in the memory of a modest supercomputer. For example, if the number of variables (n) equals 13, then constraints woud not fit in 32 GB of ram and if n=15, then the constraints would not even fit into a terabyte of memory.
> 4. “Several alternative methods for DBN structure learning were neither mentioned nor compared, such as constraint-based algorithms like PCMCI and PCMCI+...” - We managed to install and run varlingam with default settings over the few days which shows that varlingam performs substantially worse. The graphs will be added to the updated PDF.
> 5. “There are some typographical errors… ” - Thanks, we will fix this in the updated PDF.
> ## Answers to the questions:
> 1. “How can we select the parameters …” - REG is not a parameter, it denotes the regularization part of the cost function, which contains parameters $\lambda$ and $\eta$. These parameters are set to a small values so that the value of the regularization part of cost function does not change the “main objective of the optimization”. The parameters $\lambda$ and $\eta$ were fine tuned by hyperparameters search within their small magnitudes to get the fastest MIP GAP decrease.
> 2. “What is the termination condition for the proposed algorithm…” - For the purposes of the article, we have run the algorithm for a fixed time (2 hours). One could also incorporate a stop condition based on the observation of the MIP GAP, which would terminate the computation when a certain distance from the global optimum is reached.
> 3. “How does the proposed method differ from or improve upon existing techniques” - The method proposed has global convergence guarantees and a runtime readout (MIP GAP) corresponding to the distance from the optimal solution (ML estimator).
>
> We will post the updated pdf with the changes in coming days. If you could please answer our clarifications questions, we would include the changes based on these answers in the updated PDF too.

---

> > ### Author Response · Authors · 2024-11-28
> >
> > Dear reviewer, we posted an updated PDF. We fixed typographical errrors, and added comparison to VarLiNGAM.

---

> > ### Comment · Reviewer_aH4X · 2024-12-02
> >
> > Thank you for the modifications in the paper and respond to the questions. The revised version has gotten better in detail and prepared some comparisons with the varLingam algorithm.
> >
> >
> > In general, global convergence is not guaranteed for all MIP problems due to the inherent combinatorial and non-convex nature of these problems. However, specific algorithms or approaches can achieve global convergence under certain conditions or for specific types of MIP problems. For example, Branch-and-Bound/Branch-and-Cut algorithms guarantee global convergence to the optimal solution if: 1) The feasible region is bounded, and 2) The solver is allowed to explore all branches (which may require exponential time in the worst case), but for large or complex MIP problems, practical limits like time or memory constraints may prevent full convergence. So, in the proposed algorithm, I think the runtime and memory constraints are the main challenges. There is no comparison in runtime with other existing approaches.
> >
> >
> > I modified my score from 3 to 5.

---

### Official Review · Reviewer_Tekg · 2024-10-28

**Soundness:** 2
**Presentation:** 1
**Contribution:** 2
**Rating:** 3
**Confidence:** 4

**Summary:**

The paper presents an exact dynamic Bayesian network structure learning algorithm named ExDBN based on a mixed-integer quadratic program. The authors formulate the structure learning problem as a mixed-integer program and propose a new acyclicity constraint named lazy constraint. The proposed constraint helps tackle the curse of dimensionality since it does not require pre-generation in front of structure learning.

**Strengths:**

1. The paper proposes a novel acyclic constraint for score-based structure learning algorithms.
2. The authors conducted plenty of experiments to validate their proposed algorithm.

**Weaknesses:**

1. This paper contains numerous writing errors that severely affect readability.
2. Although the authors claim that their algorithm is an exact learning algorithm, there is no theoretical proof or analysis of their method.
3. There are no ablation studies to show the effectiveness of the proposed lazy constraints.
4. There is a lack of comparison with other exact BN structure learning algorithms.

**Questions:**

1. In the experiments, why is only one set of the hyperparameters of ExDBN being evaluated? How did you choose the hyperparameters for each scenario?
2. What is MIP GAP? Could you please clarify this term in more detail?

---

> ### Author Response · Authors · 2024-11-22
>
> We thank the reviewer for taking the time to evaluate our paper, we do however want to provide a strong rebuttal, since there seems to be a fundamental confusion in relation to certain key concepts that we build upon.
> 1. “This paper contains numerous writing errors that severely affect readability.” - Could you please be more specific?
> 2. “although the authors claim that their algorithm is an exact learning algorithm, there is no theoretical proof or analysis of their method.” - Using the branch and bound and cut method guarantees global convergence in general (basic theory, see here for instance: https://www.sciencedirect.com/topics/computer-science/branch-and-cut), typically at the cost of scalability. We deal with the scalability issue in the article (see Sections 2 through 4).
> 3. “There are no ablation studies to show the effectiveness of the proposed lazy constraints.” - The potential number of constraints is  equal to the number of directed cycles in a complete graph on n vertices, which is $\sum_{i=1}^n \binom{n}{i}(i-1)!$. Due to this count being very large, the examples we present would not fit in the memory of a modest supercomputer. For example, if the number of variables (n) equals 13, then constraints woud not fit in 32 GB of ram and if n=15, then the constraints would not fit into a terabyte of memory.
> 4. “There is a lack of comparison with other exact BN structure learning algorithms.” - Could you please mention some of these methods? This would allow us to make an additional comparison.
> ### Answers to questions:
> 1. “In the experiments, why is only one set of the hyperparameters of ExDBN being evaluated? How did you choose the hyperparameters for each scenario?” - The hyperparameters were chosen based on the observation of the dual gap, which is something that we may do, even when the ground truth is not known. Several candidate combinations of hyperparameters were chosen and the one with the most rapid decrease in dual gap is displayed.
> 2. “What is MIP GAP? Could you please clarify this term in more detail?” - MIP GAP stands for Mixed integer program gap, this is computed by abs[prim_sol - dual_sol]/abs[prim_sol]. This has not been clarified since it is undergraduate level knowledge. Please refer to an introductory book on integer programming https://link.springer.com/book/10.1007/978-3-319-11008-0.

---

### Official Review · Reviewer_Lb4r · 2024-11-03

**Soundness:** 2
**Presentation:** 2
**Contribution:** 3
**Rating:** 3
**Confidence:** 3

**Summary:**

The paper describes an exact score-based algorithm for inferring dynamic Bayes networks from sample data that maximizes the likelihood of the data. The dynamic Bayes network has both inter-temporal edges (guaranteed to be acyclic by virtue of all of the edges pointing forward in time) and intra-temporal edges which are assumed to be acyclic. The proposed branch-and-bound-and cut algorithm is an exact algorithm that is made compuationally feasible on larger numbers of variables than competitor algorithms by virtue of not adding all of the cyclicity constraints at the beginning of the search, but by eliminating cyclic graphs as they arise during the search. This saves time by eliminating the precomputation of a super-exponential (in the number of vertices) cyclicty constaints. The algorithm is tested on simulated data, where it generally was faster than the alternative DynoTears and had less variance in the results output. It was also applied to real data sets from biomedicine and finance.

**Strengths:**

The major strengths of the paper is that it describes an exact algorithm for searching for linear dynamic Bayesian networks with Gaussian or non-Gaussian noise, that maximizes the likelihood, and takes greatly reduced time relative to other algorithms. I did not see any technical problems with the paper. The problem was explained clearly, and the modification to previous algorithms was described clearly.

**Weaknesses:**

There were  a number of weaknesses in the simulation study and in the exposition. The simulation study did not do anything particularly unusual, but it did not use best practices.

According to the paper, "We generated data in a manner similar to that described in Zheng et al. (2018a) and Pamfil et al. ... Note that we used slightly different generator than in Pamfil et al. (2020) on which DYNOTEARS algorithm perform worse than in original article. We adapted ER and SF generators from Zheng et al. (2018a) for dynamic networks. (2020)." However, the simulations in Zheng et al. (2018a) have been justly criticized in a number of articles. It has been noted that the output of NOTEARS was very sensitive to a number of different features of the data that could be primarily due to the simulation method, rather than being generally true of real data. For example, NOTEARS is sensitive to scaling (i.e. choice of units of measurement), and that the performance of  NOTEARS declined extensively when the data was normalized (Kaiser and Sipos, 2022). It has also been suggested that the performance of continuous optimization algorithms benefit from error terms of equal variance (Ng et al. 2024), the similarity of the order of the variance or R2 of variables to their place in the causal order (Reisach et al., 2021, 2023). The paper does not specify how the variance of the error terms was generated so it is hard to tell whether these problem would  apply in their simulations. The worries about the simulations also affects the authors justification for only comparing ExDBN to DYNOTEARS (because of its superior performance over competitors), but the judgement about superior performance was based on the DYNOTEARS simulation, which is questionable. The authors also make recommendations about heuristic choices of search parameters, based upon the simulation results. They also seem to choose which search parameters to compare to DYNOTEARS under various circumstances based on earlier simulations (or they are only presenting the search parameters with the best results) based on simulations. These are all also questionable unless the generalizability of the simulation results can be confirmed.

There is a lot of emphasis on the speed of ExDBN. The authors state that their algorithm does not suffer much from the curse of dimensionality. However, while there are scattered remarks about execution times,  there are no tables or graphs of execution times.

The choice of coefficients for the intra and inter temporal edges is described but not justified. The intra temporal edge coefficients are in a fairly narrow range which is not explained.

The authors mention that "c > 0 is the maximal admissible magnitude of any weight". I am unclear about whether this means that c is input as a parameter for the search. If it is, does it affect the speed of the search significantly? How should the value of c be chosen? Presumably, one could just stick in some enormous number, but if that affects the speed or some other aspect of the algorithm that would not be a good idea. Also, they did specify a range for the linear coefficients in their simulations. Was the true value of c given to the algorithm in the simulations?

In the analysis of the financial data, they compare their algorithm to a heuristic algorithm from Ballester. They emphasize that their algorithm is not heuristic, but they do not say how long the Ballester algorithm took or how the results of the Ballester algorithm compared to their results. They also did not describe using DYNOTEARS on the finance data.

The authors compare DYNOTEARS and ExDBN on some biological data. From their description, it is hard to compare exactly how long either algorithm took, and how they compared with respect to the different evaluation scores. Also, they should explain what a duality gap is.

As the authors note, they did not test the algorithm on non-Gaussian noise.

Minor points:
The G score used in the simulations is undefined. After defining SHD and F1 scores, the authors describe two possible scores relating to the strengths of the edges - neither is identified as the G score. I think it is the one in equation (17), but it should be stated explicitly.

The graphs presenting the simulation results are a little confusing. I think that the x axis in each case is the number of variables, but this caption appears only in some scattered subset of the graphs. In their key explaining what each line and color means, they show the min and the mean  as solid and dotted black lines, even though there aren't any solid or dotted black lines in the graphs. What they mean is that whatever the color the line is, the solidity denotes whehter it is min or mean.

There is no description of the branch and bound and cut algorithm (which is referred to in the abstract as the branch and cut algorithm). There is just a reference to a Ph.D. thesis for the branch and bound algorithm, and then a description of the cut part of the algorithm. Some minimal description of the branch and bound algorithm would be helpful.

It would be helpful to have some description of the "mild" conditions for identification of W.

The authors state that "It should be noted, that naturally the worst possible value and the mean can together be used to bound the variance with respect". Respect to what?

On line 88, "manor" is mispelled.

Marcus Kaiser and Maksim Sipos. Unsuitability of NOTEARS for causal graph discovery
when dealing with dimensional quantities. Neural Processing Letters, 54:1587–1595, 2022.

Ignavier Ng, Biwei Huang, and Kun Zhang. Structure learning with continuous optimization:
A look and beyond. In Causal Learning and Reasoning, pages 71–105. PMLR,
2024.

Alexander Reisach, Christof Seiler, and Sebastian Weichwald. Beware of the simulated
DAG! causal discovery benchmarks may be easy to game. In Proceedings of the Conference
on Advances in Neural Information Processing Systems, volume 34, pages 27772–27784,
2021.

Alexander Reisach, Myriam Tami, Christof Seiler, Antoine Chambaz, and Sebastian Weichwald.
A scale-invariant sorting criterion to find a causal order in additive noise models.
In Advances in Neural Information Processing Systems, volume 36, pages 785–807, 2023.

**Questions:**

How were the variances of the error terms in the simulations chosen?
How were the ranges for the linear coefficients chosen?]
What were the exact runtimes for each of the algorithms?
Was the true value of the maximum linear coefficient bound c given to the algorithm in the simulations?
How does the choice of c affect the performance of the algorithm?
How should the value of c be chosen?

---

> ### Author Response · Authors · 2024-11-22
>
> We thank the reviewer kindly for their notes and remarks, we would like to, however, take a stance on some of the points made:
> 1. “The simulation study did not do anything particularly unusual, but it did not use best practices” - could the reviewer please specify what he means by best practices?
> 2. You suggest that the numerical evidence is unsatisfactory, namely that we have compared our solver with DYNOTEARS using their generation method. Would you please recommend another way of benchmarking the solver using ER and SF generation methods?
> We agree with your points on sensitivity to scaling and normalisation, which are valid comments which pertain to DYNOTEARS. It’s important to note that there may be several causes. As we have emphasised DYNOTEARS is a local solver and so no guarantees of global convergence implies a possibly arbitrarily bad result. Another shortcoming, which is present for both EXDBN and DYNOTEARS is the inability to consider non-linear dependencies. This stems from the use of a linear SEM model, which is the basis of both the models. Either way, the references that you mention provide only negative results and do not move the state of the art and so benchmarking vs. DYNOTEARS is still a pretty solid way to give a good idea about the overall performance.
> 3. “There is a lot of emphasis on the speed of ExDBN” - We are not aware of such a strong emphasis. We do mention that in the risk transmission example we do get the solution within 2 minutes and that we do not observe exponential slowdown in the experiments, but the main feature of the solver is the global convergence to the maximum likelihood estimator, see the last sentence of the conclusion. For most of the problems we get results of similar quality with less time. We could definitely produce a graph which would compare solution quality with running time.
> 4. “The intra temporal edge coefficients are in a fairly narrow range which is not explained.” - The generation method fails if these do not decay fast enough, detail may be added.
> 5. “The choice of coefficients for the intra and inter temporal edges is described but not justified” - This is inspired by the numerical study provided for the evaluation of DYNOTEARS
> 5. "c = 100 is the maximal admissible magnitude of any weight" - this is just a big enough number that will not affect the identification, this does not affect performance or solution.
> The minor points can all be amended by editing the document. Thank you!
>
> ### Answers to questions:
> 1. “How were the variances of the error terms in the simulations chosen?” - Variance was 1.
> How were the ranges for the linear coefficients chosen? In the benchmarks the weights of sampled ground truth graphs were sampled from [-0.4,-0.2]\cup[0.2,0.4] on inter graph edges and {-0.5, 0.5}.  These are basically default values of the used generator (adapted from DYNOTEARS). We only adjust the interval on the inter-edge graph in order to get a non-divergent series.
> 2. “What were the exact runtimes for each of the algorithms?” - The run time of ExDBN was 2 hours compared to a few seconds for DYNOTEARS. Note that this time limit was somehow artificial. For most of the problems we could have similar quality results with less time. We could definitely produce a graph which would compare solution quality with running time.
> 3. “Was the true value of the maximum linear coefficient bound c given to the algorithm in the simulations” - No, c is chosen so that it does not affect the optimization, i.e. c=100 and the bound on the edge weights found is 0.5.
> 4. “How does the choice of c affect the performance of the algorithm & How should the value of c be chosen” - We did not try to tinker with c. We did not have a reason to think that it would affect performance.
>
> We will post the updated pdf with the changes in coming days. If you could please answer our clarifications questions, we would include the changes based on these answers in the updated PDF too.

---

> > ### Comment · Reviewer_Lb4r · 2024-11-25
> > **Answers to author questions**
> >
> > I am not denying that your algorithm isn't the best. I don't think the simulation study was good evidence of that.
> >
> > 1. Setting all of the variances of the error terms to be equal makes a DAG identifiable in the linear Gaussian case (as opposed to making a Markov equivalence class identifiable if the variances are not equal. So the variances should not be set to be equal. Does varsortability  affect your results? Does standardization affect your results? Does the noise ratio affect your results? All of these are things that seem to appear in the NOTEARS simulation, which are not guaranteed to appear in real data.
> >
> > The gap in the coefficients selected is also questionable in two ways (although I recognize that it is not uncommon to leave such a gap in simulations). The gap means that all of the results are only for detection of strong edges. But even if you are interested only in the strong edges, it is not clear that the gap won't also improve the performance on the strong edges, since the existence of weak edges could make the detection of strong edges more difficult. Not leaving a gap, or seeing how the gap affected the results would be valuable information.
> >
> > The point is that the way that the data was generated improved the performance of NOTEARS, and we don't know that the same is not happening in your paper.
> >
> > 2.  Why not compare it to Demiralp and Hoover, or Bessler and Lee, or Hyvarinen, or Moneta?
> > Alessio Moneta, Nadine Chlaß, Doris Entner, and Patrik Hoyer. Causal search in structural
> > vector autoregressive models. In Proceedings of the NIPS Mini-symposium on Causality
> > in Time Series, pages 95–114, 2011.
> >
> > 3. The TL;DR at the top of your article was: "Mixed-integer formulations can used to learn dynamic Bayesian nets very fast, when valid violated inequalities are generated dynamically." You also say " It is shown, that given sufficient data, only a small amount of these constraints are actually needed to ensure the acyclicity of the resulting graph, which leads to the runtime generation of these constraints granting a large speedup."
> >
> > Information about how many variables the algorithm can handle in what time period is important information.
> >
> > 4. Ok.
> >
> > 5. Just because the simulation in NOTEARS was done that way, this is not a good justification for doing it that way. As I mentioned, the NOTEARS simulation led to a large overestimate of how well NOTEARS would perform on data that was reasonably generated in different ways.
> >
> > 6. Ok.

---

> > > ### Author Response · Authors · 2024-12-01
> > >
> > > Dear reviewer,
> > >
> > > we posted an updated version of PDF. We added an additional experiment where the noise variances are different for each variable. We tried experiments with normalized data, and noticed that the ExDBN does not perform well on such data. Therefore, we added a note to the article, that we assume that the data need to have the right scale. This somewhat limits the universal applicability of ExDBN. However, it should be noted that in many applications the data is guaranteed to be scaled properly (e.g. finance, energy market trading).
> > >
> > > We added a comparison with the additional solver VarLinGam.

---

> ### Author Response · Authors · 2024-11-25
> **Thanks again for the diligent feedback!**
>
> I would like to thank you again for the through feedback,
>
> obviously, we will not be able to amend everything by the deadline, we will, however, do our best to partially address the points that you made. In particular:
>
> 1. We will include results that compare our solver with LiNGAM. Furthermore, we will study the effects of varsortability and  standardization and hopefully feature at least one of these numerical experiments by the deadline.
>
> On the coefficient gap: you are right! We have observed that identification when considering a continuous range does make identification more difficult (the presence of weak edges as you have called it). We will use this as an idea to improve benchmarking in the future.
>
> 2. We are adding a comparison with LiNGAM (Hyvarinen) to bolster the results.
>
> 3. When talking about performance, we did not mean it globally with respect to solvers that are local in nature etc. We do however understand that it could be understood that way. We will amend this.
>
> 5. Taking the contents of "Beware of the Simulated DAG! Causal Discovery Benchmarks May Be Easy To Game" into account, we accept this point full and hopefully the new results mentioned above will move us closer to a more faithful assessment.
>
> Thanks again for your time!

---

### Official Review · Reviewer_vQzM · 2024-11-05

**Soundness:** 3
**Presentation:** 2
**Contribution:** 2
**Rating:** 5
**Confidence:** 3

**Summary:**

This paper presents ExDBN, a novel approach for learning Dynamic Bayesian Networks (DBNs) through a mixed-integer quadratic programming formulation. By employing a branch-and-cut algorithm to ensure acyclicity, the method achieves a globally convergent solution, especially effective for small and medium-sized datasets. Synthetic and application-focused benchmarks in biomedicine and finance demonstrate ExDBN's improved reliability compared to existing methods like DYNOTEARS.

**Strengths:**

- The paper proposes a mathematically rigorous approach, leveraging mixed-integer programming to enhance global convergence.
- The lazy constraint strategy for acyclicity constraints is innovative and may reduce computational load.
- The benchmark experiments, including applications in biomedicine and finance, demonstrate ExDBN’s potential practical relevance.
- Data generation is explained well, detailing both the DAG and time-series structures used, which aids reproducibility.

**Weaknesses:**

- Novelty and contribution are marginal, as the proposed method offers limited improvements over existing techniques.
- The paper includes three pages of bulky and sparsely populated figures that disrupt the flow and readability. Merging figures or including only key results would improve coherence, while additional figures could be moved to an appendix.
- There is limited comparison with other methods, potentially weakening claims about ExDBN's performance advantages.
- Results are presented without adequate explanation or interpretation, leaving readers with limited understanding of the findings' significance.
- Shortcomings and limitations of the proposed method are not discussed, which reduces transparency and leaves gaps in critical evaluation.
- The proposed method lacks solid grounding; the choice of mixed-integer programming is not fully justified, and its advantages over alternative approaches are unclear.
- For readers unfamiliar with mixed-integer programming, an introductory section on the topic would improve accessibility.
- Key results and findings are not emphasized in the abstract, reducing its effectiveness in capturing the study's contributions. It is mentioned "we show that the proposed approach turns out to produce excellent results." This is a qualitative description. I would help capture the readers attention if you quantitatively highlight how ExDBN out performs DYNOTEARS.
- Several key related works are not cited, including foundational and recent contributions, which leaves the literature review incomplete. Relevant missing citations include:
     - "Learning the Structure of Dynamic Probabilistic Networks" by Nir Friedman, Kevin Murphy, and Stuart Russell
     -  "Learning Dynamic Bayesian Networks from Data: Foundations, First Principles and Numerical Comparisons" by Vyacheslav Kungurtsev et al.
     -  "GRACE-C: Generalized Rate Agnostic Causal Estimation via Constraints" by Mohammadsajad Abavisani et al.
     -  "Divide-and-Conquer Strategy for Large-Scale Dynamic Bayesian Network Structure Learning" by Hui Ouyang et al.

**Questions:**

- How does ExDBN perform when the assumptions regarding data distribution or noise are violated?
How does ExDBN handle missing data or noisy measurements, which are common in practical applications?
- Could you elaborate on the computational requirements of ExDBN compared to DYNOTEARS, especially for high-dimensional datasets?
- Have you considered applying ExDBN to datasets with nonlinear dependencies to test the flexibility of the mixed-integer programming formulation?
- What strategies might practitioners use to select regularization parameters in the absence of known ground truth?

---

> ### Author Response · Authors · 2024-11-22
>
> We thank the reviewer for their insightful comments pertaining to the article, however, we would like to offer the following rebuttal:
>
> 1. “Novelty and contribution are marginal, as the proposed method offers limited improvements over existing techniques.” - The utility of a novel approach may be measured in its performance compared to other solvers. Here we have demonstrated a non-trivial improvement in many instances, see Figures 1 through 3. A further advantage of this method is its simplicity.
> 2. “The paper includes three pages of bulky and sparsely populated figures that disrupt the flow and readability. Merging figures or including only key results would improve coherence, while additional figures could be moved to an appendix.” - We would be happy to reformat to improve the flow.
> 3. “There is limited comparison with other methods, potentially weakening claims about ExDBN's performance advantages.” - Could you please recommend other solvers to compare to? We have done further performance tests comparing to lingam, which we will attach.
> 4. “Results are presented without adequate explanation or interpretation, leaving readers with limited understanding of the findings' significance.” - We have provided standard contextualization and benchmarking associated with this type of identification. In particular, comparing graphs in terms of G and F scores and SHD in conjunction with SF and ER generation methods is a common practice.
> 5. “Shortcomings and limitations of the proposed method are not discussed, which reduces transparency and leaves gaps in critical evaluation.” - It is true that the shortcomings are not listed in a verbose section, but the key aspects are discussed in the conclusion. Furthermore, the shortcomings could be summarised as follows: we do not scale as far as local solvers (like DYNOTEARS), but we have shown that on mid sized instances (the ones presented in the numerics section), we are able to outperform them. We are happy to amend such a discussion.
> 6. “The proposed method lacks solid grounding; the choice of mixed-integer programming is not fully justified, and its advantages over alternative approaches are unclear.” - as explained in the previous point, mixed integer programming solvers such as Gurobi have a guarantee of convergence to the global minimizer, by design. Additionally, one has a duality gap at their disposal, which bounds the error from above at any time prior to convergence. As long as we can agree that the formulation of the problem is correct, the use of integer programming is an option. This is shown to be viable if one is able to avoid generating exponentially many constraints, which we do.
> 7. “For readers unfamiliar with mixed-integer programming, an introductory section on the topic would improve accessibility.” - We will include a short paragraph that defines a mixed-integer program along with some basic references that contain detailed introductions.
> “Key results and findings are not emphasised in the abstract….” - We will amend the abstract.
> “Several key related works are not cited, ” - This can easily be amended.
> ## Answers:
> 1. “How does ExDBN perform when the assumptions regarding data distribution or noise are violated? How does ExDBN handle missing data or noisy measurements, which are common in practical applications?” - On a theoretical level, there should be no problem in identifying some types of non-gaussian noise as the model is based on a ML estimate valid for a wider range of noises (see end of Section 2). No robust statistical approach, which would handle outliers, has been implemented yet.
> 2. “Could you elaborate on the computational requirements of ExDBN compared to DYNOTEARS, especially for high-dimensional datasets?” - The run time of ExDBN was 2 hours compared to a few seconds for DYNOTEARS. Note that this time limit was somehow artificial. For most of the problems we get similar quality results with less time. We could definitely produce a graph which would compare solution quality with running time.
> 3. “Have you considered applying ExDBN to datasets with nonlinear dependencies to test the flexibility of the mixed-integer programming formulation?” - ExDBN is based on a linear SEM model, which does not assume any non-linear interaction and thus is not suitable for capturing them. But we think the method could generalise to quadratic interactions which will be subject to future research.
> 4. “What strategies might practitioners use to select regularisation parameters in the absence of known ground truth?” - One could use the dual gap decrease rate, which is available during the solution of the MIQP; a better rate would be favourable in this case. The regularisation should be kept small in order to ensure that the problem meaning does not change.
>
> We will post the updated pdf with the changes in coming days. If you could please answer our clarifications questions, we would include the changes based on these answers in the updated PDF too.

---

> > ### Author Response · Authors · 2024-11-28
> >
> > Dear reviewer, we posted an updated PDF. We updated abstract, added citations, included a short paragraph about MIP, reformatted graphs, discussed some shortcomings of our method.

---

> > > ### Comment · Reviewer_vQzM · 2024-12-02
> > > **Thank you for revisions**
> > >
> > > I thank the authors for addressing the comments. I think the paper looks much better now. I am increasing my score to 5. However, I still think the paper benefits from comparison with more methods.

---

### Meta-Review · Area_Chair_8FtA · 2024-12-21

**Metareview:**

This paper proposes an exact method for learning dynamics BNs. Two reviewers increased their score during discussion, but still lean towards reject. The other two reviewers favor outright reject. Based on the discussion, it seems there are some useful ideas for the community here, but the paper needs another round of revision to address reviewer concerns before it can be accepted.

**Additional Comments On Reviewer Discussion:**

Two reviewers increased their score, and there was a back and forth with another reviewer who was not convinced.

---

### Decision · Program_Chairs · 2025-01-22

Reject